**Monitoring aseismic creep trend in Ismetpasa and Destek segments throughout the NAF with a**
**large scale GPS network**
Hasan Hakan Yavaşoğlu[1,*], Mehmet Nurullah Alkan[2], Serdar Bilgi[1], Öykü Alkan[3]
[1] ITU, Dept. of Geomatics Engineering, Maslak, Istanbul, Turkey.
[2] Hitit University, Osmancık MYO, 19030, Corum, Turkey
[3] ITU, Graduate School of Science Engineering and Technology, Maslak, Istanbul, Turkey
**Abstract**
North Anatolian Fault Zone (NAFZ) is an intersection area between Anatolian and Eurasian
plates. Arabian plate, which squeezes the Anatolian plate from the south between Eurasian plate and
itself is also responsible for this formation. This tectonic motion causes Anatolian plate to move
westwards with almost a 20 mm/year velocity which has caused destructive earthquakes in the history.
Block boundaries that form the faults are generally locked to the bottom of seismogenic layer because
of the friction between blocks, and responsible for these discharges. However, there are also some
unique events observed around the world, which may cause partially or fully free slipping faults. This
phenomenon is called "aseismic creep", and may occur through the entire seismogenic zone or at least
to some depths. Additionally, it is a rare event in the world located in two reported segments along
the North Anatolian Fault (NAF) which are Ismetpasa and Destek.
In this study, we established GPS networks covering those segments and made three
campaigns between 2014-2016. Considering the long term geodetic movements of the blocks
(Anatolian and Eurasian plates),  surface velocities and fault parameters are calculated. The results of
the model indicate that aseismic creep still continues to some rates of 13.2±3.3 mm/year at Ismetpasa
and 9.6±3.1 mm/year at Destek. Additionally, aseismic creep behavior is limited to some depths and
decays linearly to the bottom of the seismogenic layer at both segments. This study suggests that this
aseismic creep behavior will not prevent a medium-large scale earthquake in the long term.
**Key words:** NAFZ, aseismic creep, GPS, block modelling
**Introduction**
Fault zones all around the world are formed by the tectonic plate motions and is a natural
boundary between blocks. They are generally locked to the bottom of seismogenic layer and cannot
slip freely compared to the velocities within the blocks because of the friction between rocks.
Therefore, movement in these regions generally minimal and causes earthquakes when the motion of

*Corresponding author:
(HH Yavaşoğlu), yavasoglu@itu.edu.tr

the blocks overrides the friction force. After discharge (earthquake), faults begin to accumulate strain
and this cycle continues until the next earthquake (Reid 1910, Yavaşoğlu 2011).
NAF(North Anatolian Fault) is a tectonic plate boundary between Anatolian and Eurasian
plates. It slowly moves ~20 mm/year to the west by the overthrusting Arabian plate from the south
and compresses the plate motion with the help of a massive Eurasian plate in the north. Those tectonic
forces constitute North Anatolian Fault, which lies between Karliova triple junction from the east to
the Aegean Sea to the west for almost 1200 km long. The width of the fault trace ranges between 100
m to 10 km.  Anatolian plate moves 20-25 mm/year to the west relative to the Eurasian plate. There
are velocity variations  along the fault that is, west region moves faster than the eastern part and is a
right-lateral strike slip fault (Fig. 1) (Ketin 1969-1976, McClusky et al. 2000, Cakir et al. 2005, Şengör et
al. 2005, Reilinger et al. 2006, Yavaşoğlu et al. 2011, Bohnhoff et al. 2016).

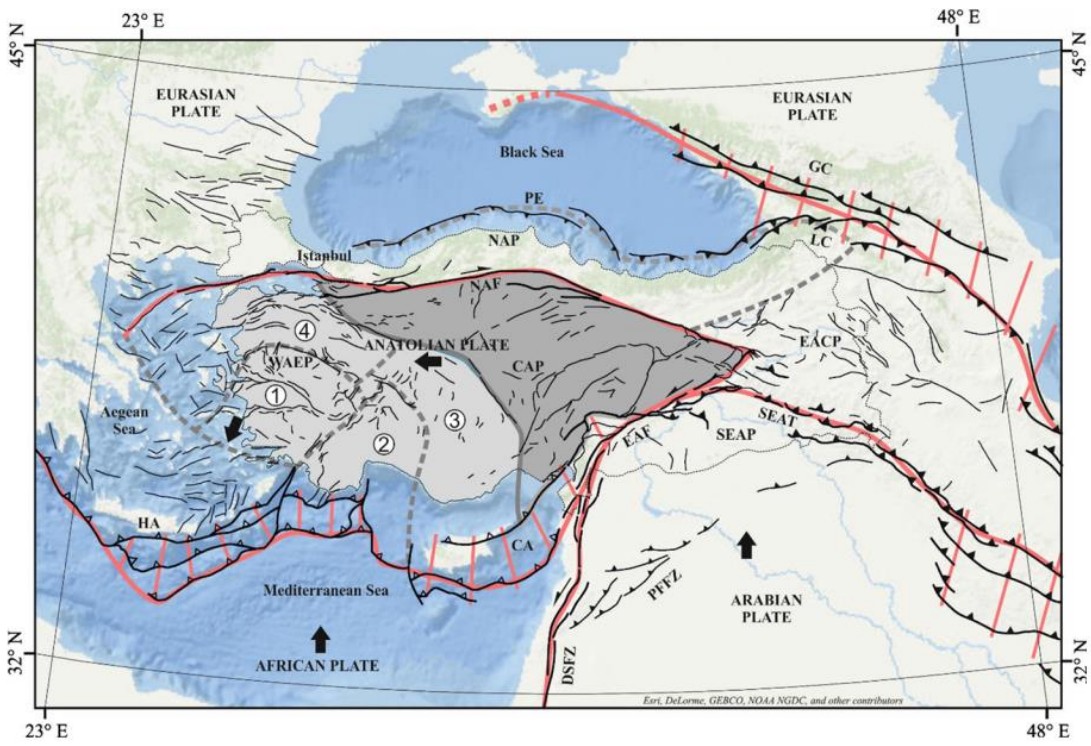


**Figure 1.** Formation of the North Anatolian Fault and interacting tectonic plates (from Emre et al.
2018). Anatolian plate moves westwards due to African and Arabian plates overthrusting. (1)West
Anatolian graben systems, (2) Outer Isparta Angle, (3) Inner Isparta Angle, and (4) Northwest
Anatolia transition zone. The original version of the figure is available in Emre et al. 2018.

Earthquake mechanisms might have different characteristics in some regions. Faults may move
freely without an earthquake and this motion reported at some unique places like Hayward fault
(Schmidt et al. 2005), the Superstition Hills fault(Wei et al. 2011) and Ismetpasa segments (Cakir et al.
2012) which can be observed from the surface(Ambraseys 1970, Yavasoglu et al. 2015). This
phenomenon is called "aseismic creep" and may occur in two different ways. If the creep takes place
to the bottom of seismogenic layer and the surface velocities are equal or close to the long-term
tectonic velocities, there will not be enough strain accumulation for a large scale earthquake (Şaroğlu
ve Barka 1995, Cakir et al. 2005). On the other hand, if that free motion is not observed to the bottom
of the seismogenic layer or observed surface velocities are smaller than the tectonic velocities, strain
will accumulate to a final earthquake (Fig. 2) (Karabacak et al. 2011, Ozener et al. 2013, Yavasoglu et
al. 2015). Also, aseismic creep in a region may occur continuously or fade out after some period
(Kutoglu et al. 2010).

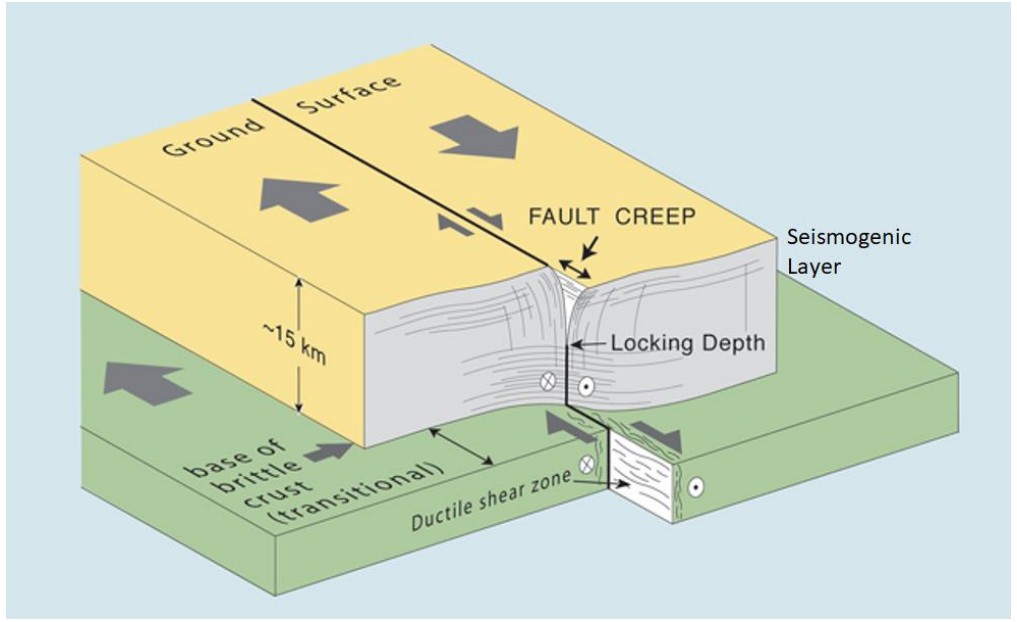


**Figure 2.** Aseismic creep structure in a fault zone. Fault may slip freely to some depths and locked
after to the bottom (URL-1).

NAF reported to have segments which show aseismic creep since 1970 at Ismetpasa and with
a more recent discovery, Destek (Ambraseys 1970, Karabacak et al. 2011). Aseismic creep at the
Ismetpasa is reported to occur along ~70-80 km, from Bayramoren (east) to the Gerede (west) (Fig. 3).
It was discovered at the wall of the Ismetpasa train station at 1970 and several minor and large scale
studies monitored the area since then (Table 1). That segment hosted three destructive earthquakes
(1943 Tosya $M_w$=7.2, 1944 Gerede $M_w$=7.2, 1951 Kursunlu $M_w$=6.9) that may have triggered or affected
the creep (Şaroğlu ve Barka 1995, Cakir et al. 2005, Karabacak et al. 2011, Kaneko et al. 2013) (Fig. 4).

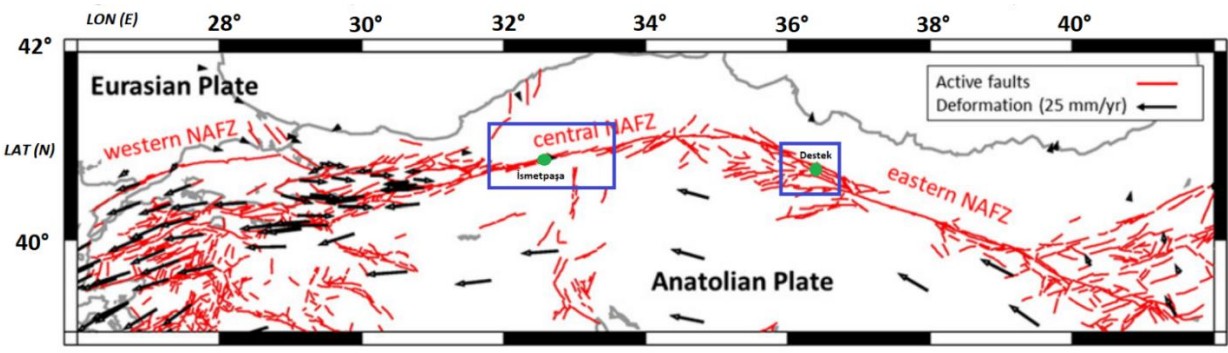

**Figure 3.** Active fault segments on the North Anatolian Fault (NAF). Blue rectangles define Ismetpasa and Destek segments from west to east, respectively (after Bohnhoff et al. 2016).

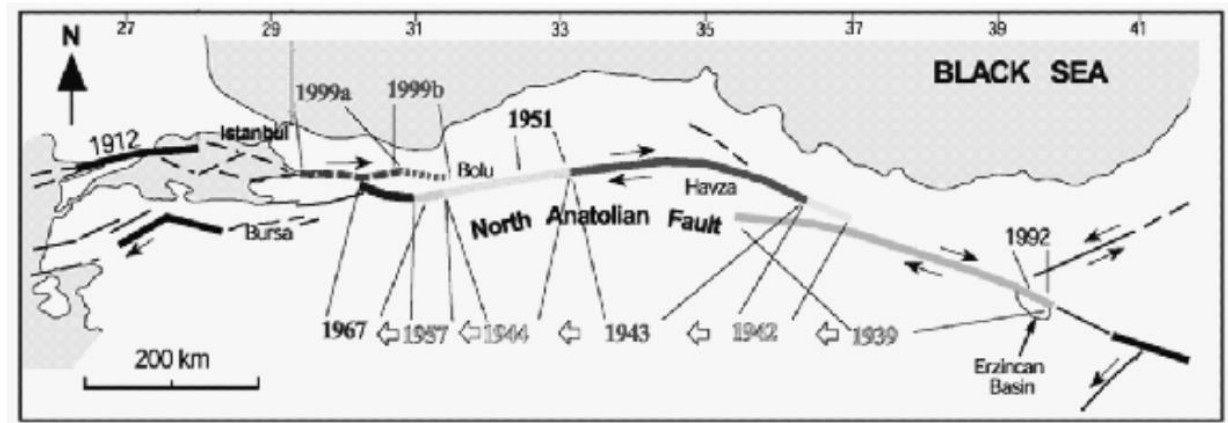

**Figure 4.** Earthquakes on the North Anatolian Fault between 1939-1999. Both 1943 and 1944 earthquakes suspected to have influence on the creeping phenomena (from Kutoglu et al. 2010).

In addition, creep at the Destek segment reported at 2003 on a field trip around the region. Unlike the Ismetpasa segment, number of studies at this segment is just a few, and also the length of this segment is unclear. 1943 Tosya earthquake, which is reportedly the biggest earthquake in the segment, affected this area (Karabacak et al. 2011) (Table 2).

All the studies around those segments indicate the continuity of creep but the results are inconsistent and cannot clearly refer whether that event has an increasing trend or not. Most of the researches (Ambraseys 1970, Aytun 1982, Eren 1984, Altay and Sav 1991, Deniz et al. 1993, Kutoglu et al. 2008&2009&2013, Karabacak et al. 2011, Ozener et al. 2013, Bilham et al. 2016) generally are on a micro-scale and focused on the Ismetpasa or a network near this village with geodetic methods, while others on a macro-scale with InSAR (Deguchi 2011, Fialko et al. 2011, Köksal 2011, Kaneko et al. 2013, Cetin et al. 2014, Kutoglu et al. 2013) which needs a ground truth (Fig 5&6).

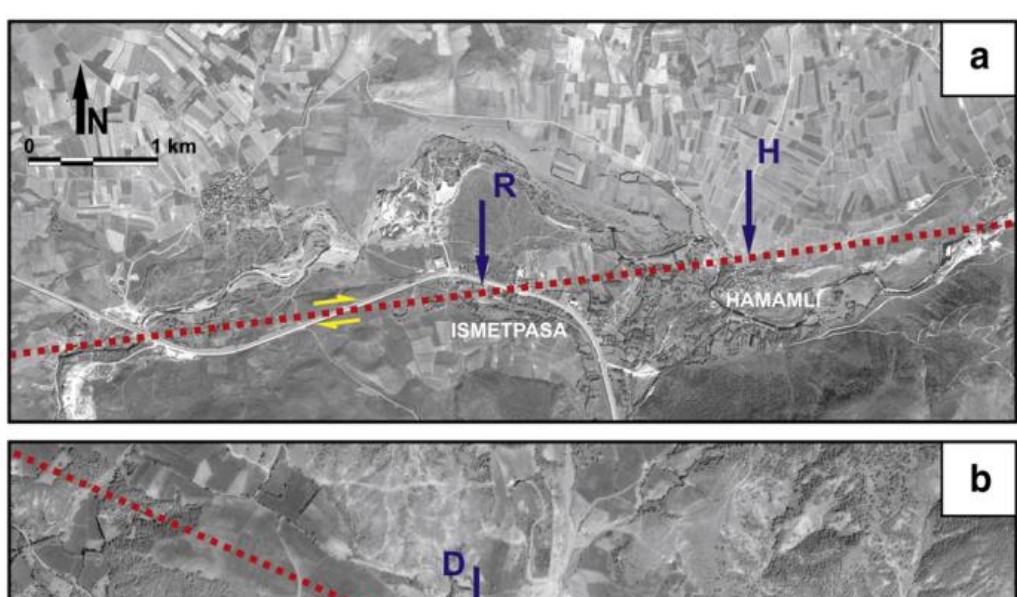

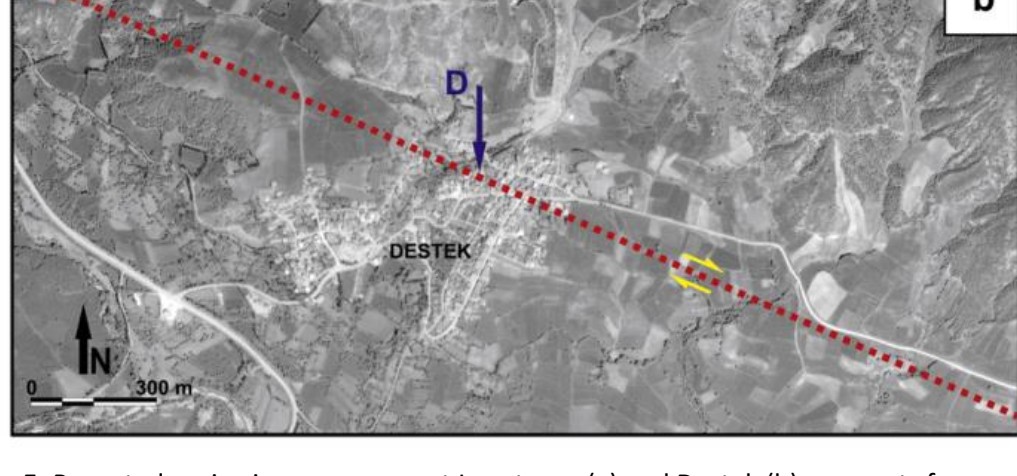

**Figure 5.** Reported aseismic creep zones at Ismetpasa (a) and Destek (b) segments from a recent study. (a) "R" shows creep observed at the wall at the Ismetpasa train station, and "H" shows the creep at Hamamli village. (b) "D" represents the reported creep at Destek town (from Karabacak et al. 2011).

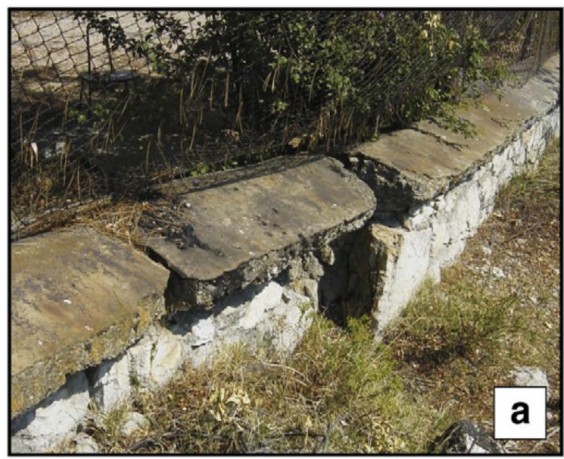
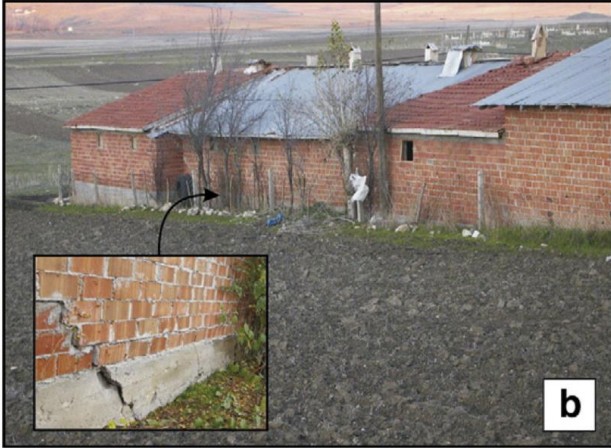

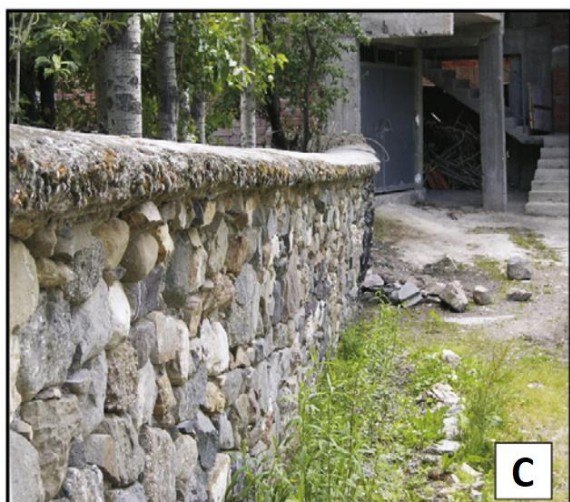

**Figure 6.** Aseismic creep sites (a)at Ismetpasa railway station, and (b) damaged brick-wall at Hamamlı village close to Ismetpasa. (c) Out-bended wall at Destek village (from Karabacak et al. 2011).

Those results cannot reveal the creep trend clearly. In addition, a ground network is required
to exhibit the fault characteristics clearly along the segments. For this reason, we established a ground
network forming profiles around those segments and made three observations annually from 2014 to
2016.

Table 1. Studies and their results to observe aseismic creep at the Ismetpasa segment between 1970-2016.

| Study | Creep rate(cm/year) | Years covered | Method |
|---|---|---|---|
| Ambraseys(1970) | 2.0 ± 0.6 | 1957-1969 | Wall offset measurements |
| Aytun(1982) | 1.10 ± 0.11 | 1969-1978 | Doppler |
| Eren(1984) | 1.00 ± 0.40 | 1972-1982 | Trilateration |
| Deniz et al.(1993) | 0.93 ± 0.07 | 1982-1992 | Trilateration |
| Cakir et al.(2005) | 0.80 ± 0.30 | 1992-2000 | InSAR |
| Kutoglu&Akcin(2006) | 0.78 ± 0.05 | 1992-2002 | GPS |
| Kutoglu et al.(2008) | 1.20 ± 0.11 | 2002-2007 | GPS |
| Kutoglu et al.(2010) | 1.51 ± 0.41 | 2007-2008 | GPS |
| Karabacak et al.(2011) [1.region] | 0.84 ± 0.40 | 2007-2009 | LIDAR |
| Karabacak et al.(2011) [2. region] | 0.96 ± 0.40 | 2007-2009 | LIDAR |
| Deguchi(2011) | 1.4 | 2007-2011 | PALSAR |
| Fialko et al.(2011) | 1.0 | 2007-2010 | PALSAR |
| Ozener et al.(2013) | 0.76 ± 0.10 | 2005-2011 | GPS |
| Köksal(2011) | 1.57 ± 0.20 | 2007-2010 | DInSAR |
| Görmüş(2011) | 1.30 ± 0.39 | 2008-2010 | GPS |
| Kaneko et al.(2013) | 0.9 ± 0.2 | 2007-2011 | InSAR |
| Cetin et al.(2014) | 0.8 ± 0.2 | 2003-2010 | InSAR(PSI) |
| Altay and Sav(1991) | 0.76 ± 0.1 | 1982-1991 | Kripmetre |
| Kutoglu et al.(2013) | 1.3 ± 0.2 | 2008-2010 | GPS |
| Kutoglu et al.(2013) | 1.25 ± 0.2 | 2007-2010 | InSAR |
| Ambraseys(1970) - Bilham et al.(2016) revision | 1.04 ± 0.04 | 1957-1969 | Revaluation of photographs |
| Aytun(1982) | 1.50 | 1957-1969 | Revaluation of photographs |
| Aytun(1982) – Bilham et al.(2016) revision | 1.045 ± 0.035 | 1957-1969 | Revaluation of photographs |
| Bilham et al.(2016) | 0.61 ± 0.02 | 2014-2016 | Creepmeter |

Table 2. Studies and their results to observe aseismic creep at the Destek segment.

| Study | Creep rate (cm/year) | Years covered | Method |
|---|---|---|---|
| Karabacak et al.(2011) | 0.66 ± 0.40 | 2007-2009 | LIDAR |
| Fraser et al.(2009) | 0.6 | 2009 | Trench study |

**Network Design Around the Creeping Segments**

Designing a monitoring network around tectonic structures is always related to the geological characteristics and fault geometry, which includes the locking depth and earthquake related motions (coseismic movements) through the fault. Previous studies indicate that the velocities for the stations distant from the fault plane can be used to derive long-term plate velocities, while nearby station velocities are suitable to detect the locking depth of a fault (Taskin et al. 2003, Halıcıoğlu et al. 2009). In addition, velocities of the observation stations gradually decrease when their locations approach to the fault plane. Another factor is the number of the stations and this is related to the fault length and width, but the station locations perpendicular to the fault plane must not exceed the ($\pm 1/\sqrt{3}$) of the

locking depth. Also, several research specify this limit to the double of the depth (Taskin et al. 2003,
Kutoglu and Akcin 2006, Kutoğlu et al. 2009, Halıcıoğlu et al. 2009, Poyraz et al. 2011, Bohnhoff et al.
2016). For this purpose, the following equation is used in general to obtain to proper distances of the
observation stations from the fault plane:

$$V(x) = \frac{V_T}{\pi} \arctan(\frac{x}{D}) \tag{1}$$

where:
- $V$ : Fault parallel velocity
- $V_T$ : Long term tectonic plate velocity
- $x$ : Distance to fault plane
- $D$ : Locking depth of the fault (Halıcıoğlu et al. 2009).
Location of the stations may vary according to the geological surface elements, but they are
generally established on the both sides of the fault to form a profile on each block to obtain surface
velocities (Yavasoglu et al. 2015).
Geologic structure at the tectonic block boundaries and fault plane geometry also affects the
tectonic behaviour. To better understand this mechanism, established network around the fault zone
is observed with different techniques periodically or continuously. The variation of the observations
are clues to detect those amplitudes, and GPS is the most common technique for that kind of studies.
This technique is very effective and efficient to collect data from ground stations established around
the faults (Poyraz et al. 2011, Aladoğan et al. 2017).
Profiles intersect with fault plane vertically are used to estimate the locking depth. However,
in such regions like Ismetpasa and Destek, there is an additional locking depth deduced from the
previous studies, which indicates that the creeping layer of the seismogenic zone does not reach to
the bottom, but around 5-7 km depth in those areas (Kaneko et al. 2013, Ozener et al. 2013, Cetin et
al. 2014, Bilham et al. 2016, Rousset et al. 2016). For this reason, aseismic layer's attenuation depth is
another crucial element to understand the creeping mechanism (Fig 2). Also considering the 5-7 km
depth value with the *Eq.1*, station locations are chosen as 3 and 10 km on the both sides of the fault
forming profiles, while NAF general locking depth is around 15 km (McClusky et al. 2000, Poyraz et al.
2011, Bohnhoff et al. 2016).
Before the 3 epochs of observations, a network was planned forming 4 profiles at the
Ismetpasa, and 1 profile at the Destek segments and including surrounding continuous GPS
stations(Real Time Kinematic Continuously Operating Reference Stations-RTK CORS) (Fig 7). Aim of this
study was to monitor this network periodically to calculate the velocity field with combining the results
with CORS station velocities and estimate the creep ratio within the Ismetpasa and Destek segments
(Yavasoglu et al. 2015).

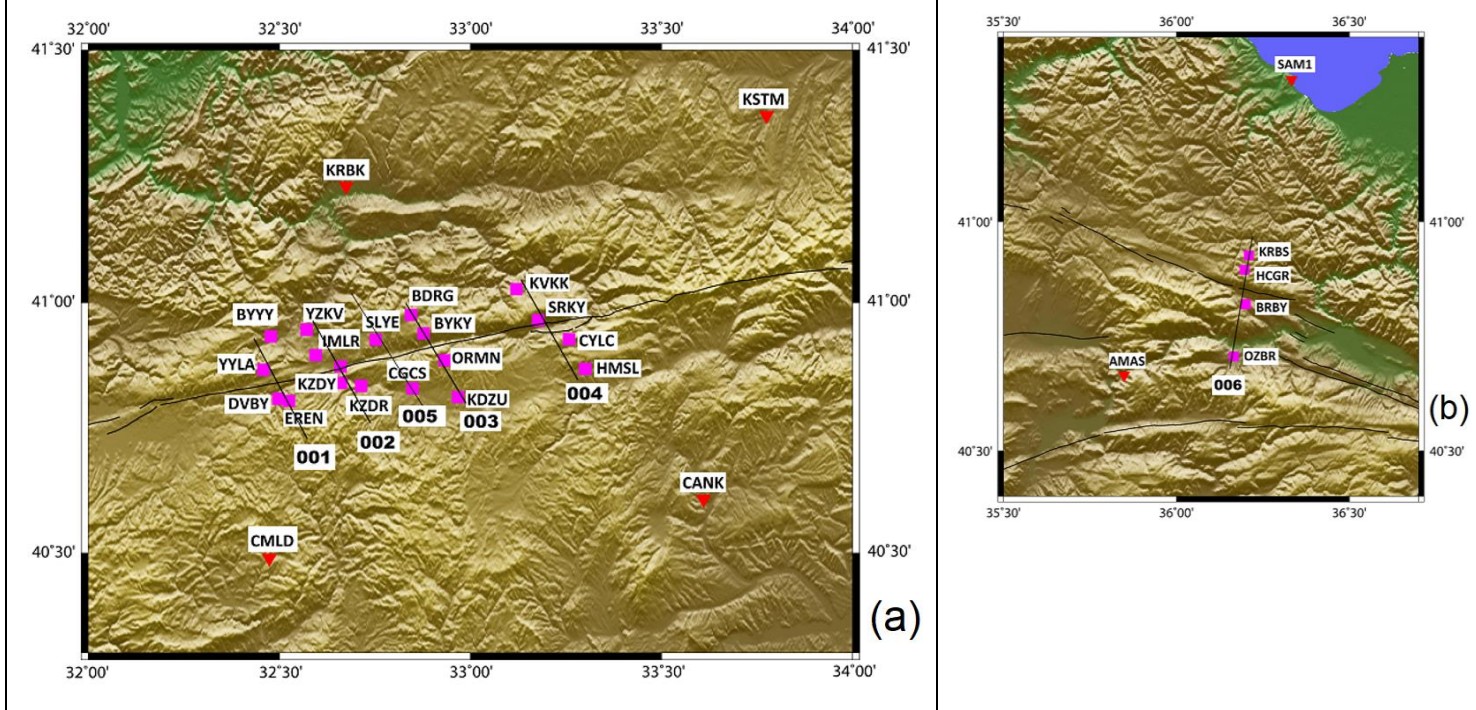

**Figure 7.** Planned profiles and campaign GPS stations(pink) at Ismetpasa(a) on the left and Destek(b)
on the right. Profiles 001-004 planned and established on the Ismetpasa segment, and profile 005
added to the network using two suitable stations. Profile 006 is on Destek segment. Fault traces on
the south of profile 006 are secondary faults. Other continuous GPS sites (RTK CORS) shown in
red(after Yavasoglu et al. 2015).
While establishing the network, first consideration for 3 and 10 km on the both sides of the
fault generally occurred, but some minor changes took place according to the geological structure of
the area. In addition, another profile between the 2$^{nd}$ and 3$^{rd}$ profiles formed with the suitable location
of two unplanned stations. Finally, there are 5 profiles within ~70 km along the Ismetpasa and 1 profile
along the Destek.
Observations are completed around the July and August for 3 years using relative geolocation
based on carrier phase observations with GPS technique (Table 3). Force centering equipment and GPS
masts were used when necessary. First campaign was on the 235-238 and 241 GPS days in 2014,
second was on 215-221 GPS days in 2015, and the last one was between 210-220 GPS days in 2016.
**Table 3.** Campaign stations, their locations and facility types.

| Profile number | Station ID | Site | Latitude (°) | Longitude (°) | Type of facility |
|---|---|---|---|---|---|
| **001** | BYYY | Büyükyayalar | 40.49 | 32.48 | Bronze |
| | YYLA | Yayla Village | 41.45 | 31.78 | Bronze |
| | DVBY | Davutbeyli Village | 39.43 | 32.50 | Bronze |
| | EREN | Elören Village | 40.81 | 32.50 | Bronze |
| **002** | YZKV | Yazıkavak Village | 40.80 | 32.53 | Bronze |
| | IMLR | İmanlar Village | 40.95 | 32.57 | Bronze |
| | HMMP | Hamamlı Village | 40.90 | 32.60 | Pillar |
| | KZDR | Kuzdere Village | 41.23 | 32.68 | Pillar |
| **005 (intermediate)** | SLYE | Kapaklı Village | 41.85 | 32.72 | Pillar |
| | CGCS | D100 wayside | 39.86 | 32.85 | Pillar |
| **003** | BDRG | Boduroğlu Village | 39.89 | 32.76 | Bronze |
| | BYKY | Beyköy Village | 40.83 | 32.85 | Pillar |
| | ORMN | Forest | 40.94 | 32.86 | Bronze |
| | KDZU | Kadıözü Village | 40.88 | 32.93 | Pillar |
| **004** | KVKK | Kavak Village | 40.81 | 32.97 | Bronze |
| | SRKY | Sarıkaya Village | 41.03 | 33.12 | Bronze |
| | CYLC | Çaylıca Village | 40.97 | 33.18 | Bronze |
| | HMSL | Hacımusla Village | 40.93 | 33.26 | Pillar |
| **006** | KRBS | Korubaşı Village | 40.82 | 36.20 | Bronze |
| | HCGR | Hacıgeriç Village | 40.71 | 36.17 | Bronze |
| | BRBY | Borabay | 40.90 | 36.20 | Pillar |
| | OZBR | Özbaraklı Village | 39.66 | 35.87 | Pillar |

After the first campaign, KZDY station was damaged and removed from rest of the project. Raw
data collected for a minimum of 8 hours at each station for the rest of the project and evaluated with
GAMIT/GLOBK software (Herring et al. 2015a, 2015b) at first, then the results used as input to block
modelling software TDEFNODE (McCaffrey 2002, 2009). A total of 63 stations (22 campaign, 30
surrounding RTK CORS, 11 IGS) are used in this network to monitor Ismetpasa and Destek segments
and the remaining region between them (Table 4).
**Table 4.** Continuous GPS(RTK CORS) stations and their locations.

| Station ID | Province | Station ID | Province | Station ID | Province |
|---|---|---|---|---|---|
| **AKDG** | Yozgat | **FASA** | Ordu | **RDIY** | Tokat |
| **AMAS** | Amasya | **GIRS** | Giresun | **SAM1** | Samsun |
| **ANRK** | Ankara | **HEND** | Sakarya | **SIH1** | Eskişehir |
| **BILE** | Bilecik | **HYMN** | Ankara | **SINP** | Sinop |
| **BOLU** | Bolu | **IZMT** | İzmit | **SIVS** | Sivas |
| **BOYT** | Sinop | **KKAL** | Kırıkkale | **SSEH** | Sivas |
| **CANK** | Çankırı | **KRBK** | Karabük | **SUNL** | Çorum |
| **CMLD** | Ankara | **KSTM** | Kastamonu | **TOK1** | Tokat |
| **CORU** | Çorum | **KURU** | Bartın | **VEZI** | Samsun |
| **ESKS** | Eskişehir | **NAHA** | Ankara | **ZONG** | Zonguldak |


**GPS Data Evaluation**

GPS data for cGPS and IGS stations' data processed at campaign observation dates. In addition,

observations for those stations during January (for 7 days) included at the GAMIT/GLOBK step to
increase the stabilization of the designed networks.

The networks linked to the ITRF 2008 global coordinate system by using surrounding IGS sites

(Table 5) (Yavaşoglu et al. 2011, Herring et al. 2015a, 2015b). After the transformation with GLOBK,
the root mean square (rms) of the stations was only 0.7 mm/year.

**Table 5.** IGS stations defined in the site.defaults file of GAMIT to constitute reference frame (*
indicates stations selected for GLOBK stabilization)

| Station ID | City/Country |
| --- | --- |
| ANKR | Ankara/Turkey |
| BUCU* | Bucharest/Romania |
| CRAO* | Simeiz/Ukraine |
| MATE* | Metara/Italy |
| ONSA* | Onsala/Switzerland |
| SOFI* | Sofia/Bulgaria |
| TEHN* | Tehran/Iran |
| TELA | Tel Aviv/Israel |
| TUBI | Kocaeli/Turkey |
| WZTR* | Koetzting/Germany |
| ZECK* | Zelenchukskaya/Russia |

Results show that the velocity of the stations located on the Anatolian plate are ranging from

15  to 20 mm/year (Fig 8), which is similar with the previous studies (McClusky et al. 2000, Reilinger et
al. 2006, Yavaşoglu et al. 2011).

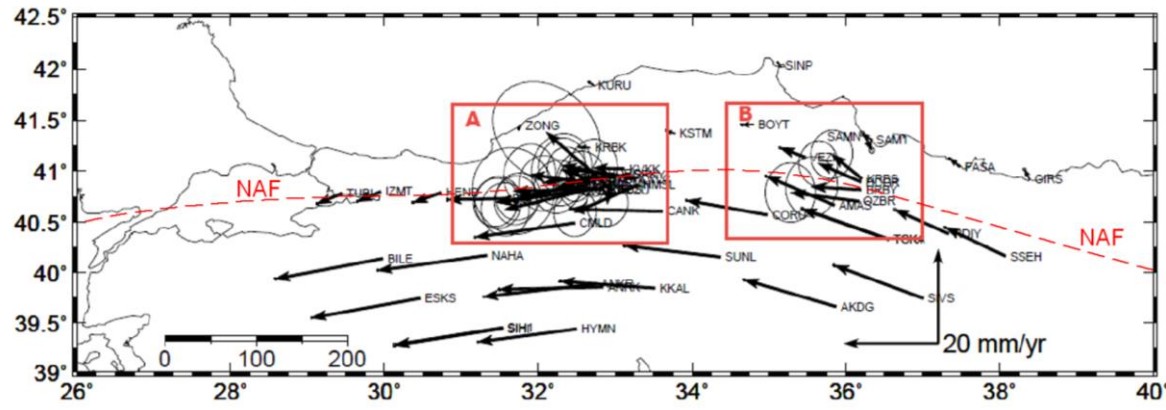


**Figure 8.** GLOBK results for station velocities relative to fixed Eurasian plate. (A) includes the
Ismetpasa segment, and Destek segment is inside (B). Dashed lines represent the fault trace of North
Anatolian Fault (NAF). Velocities at the north of the NAF are very small as expected, where south
velocities indicate the westward motion of the Anatolian plate (after Aladoğan 2017).
The GLOBK results for all of the station velocities are used as input for block modelling to
predict the aseismic creep ratio within fault plane in the predefined segments (Table 6, Fig.9).
**Table 6.** All cGPS and campaign point velocities and location errors (uncertainties) when Eurasian
plate selected as fixed.

| Station ID | Velocity(mm/yr) | | Error | | Station ID | Velocity(mm/yr) | | Error | |
|---|---|---|---|---|---|---|---|---|---|
| | $V_{EAST}$ | $V_{NORTH}$ | $V_{EAST}$ | $V_{NORTH}$ | | $V_{EAST}$ | $V_{NORTH}$ | $V_{EAST}$ | $V_{NORTH}$ |
| AKDG | -19.5 | 5.7 | 0.1 | 0.1 | KDZU | -14.1 | 12.3 | 4.6 | 4.4 |
| AMAS | -14.5 | 6.2 | 0.1 | 0.1 | KKAL | -20.1 | 1.5 | 0.1 | 0.1 |
| ANRK | -22.1 | -0.5 | 0.1 | 0.1 | KRBK | -2.3 | 0.1 | 0.1 | 0.1 |
| BDRG | -7.8 | 1.1 | 1.7 | 1.9 | KRBS | -6.4 | 5.2 | 1.8 | 2.1 |
| BILE | -22.8 | -4.3 | 0.1 | 0.1 | KSTM | -1.9 | 0.6 | 0.1 | 0.1 |
| BOLU | -12.8 | -0.2 | 0.1 | 0.1 | KURU | -0.9 | 0.5 | 0.1 | 0.1 |
| BOYT | -2.5 | -0.1 | 0.1 | 0.1 | KVKK | -6.6 | 0.2 | 2.1 | 2.5 |
| BRBY | -10.6 | 0.6 | 2.3 | 2.6 | KZDR | -18.7 | -4.5 | 2.1 | 2.3 |
| BYKY | -6.1 | -0.7 | 1.5 | 1.8 | NAHA | -23.1 | -3.2 | 0.1 | 0.1 |
| BYYY | -6.8 | -1.0 | 2.1 | 2.4 | ORMN | -0.6 | -4.4 | 1.8 | 2.0 |
| CANK | -19.4 | 0.5 | 0.1 | 0.1 | OZBR | -14.4 | 1.8 | 2.2 | 2.6 |
| CGCS | -19.2 | -0.4 | 3.5 | 3.7 | RDIY | -11.4 | 5.1 | 0.1 | 0.1 |
| CMLD | -21.1 | -3.0 | 0.1 | 0.1 | SAM1 | -1.9 | 1.3 | 0.2 | 0.2 |
| CORU | -17.2 | 3.1 | 0.1 | 0.1 | SAMN | 1.3 | -3.0 | 0.2 | 0.2 |
| CYLC | -15.5 | 2.8 | 2.0 | 2.4 | SIH1 | -22.8 | -3.6 | 0.1 | 0.2 |
| DVBY | -16.6 | -2.5 | 2.0 | 2.3 | SIHI | -22.8 | -3.6 | 0.1 | 0.2 |
| EREN | -17.6 | -2.3 | 1.9 | 2.1 | SINP | -0.7 | 0.5 | 0.1 | 0.1 |
| ESKS | -23.1 | -4.2 | 0.1 | 0.1 | SIVS | -18.8 | 7.0 | 0.1 | 0.1 |
| FASA | -2.2 | 1.8 | 0.1 | 0.1 | SLYE | -8.2 | -1.7 | 2.0 | 2.3 |
| GIRS | -1.0 | 2.1 | 0.1 | 0.1 | SRKY | -10.1 | -1.1 | 2.1 | 2.5 |
| HCGR | -9.1 | 3.9 | 1.7 | 1.9 | SSEH | -12.8 | 6.1 | 0.1 | 0.1 |
| HEND | -6.0 | -2.2 | 0.1 | 0.1 | SUNL | -20.4 | 2.4 | 0.1 | 0.1 |
| HMMP | -14.9 | -2.5 | 2.0 | 2.0 | TOK1 | -18.4 | 6.4 | 0.1 | 0.1 |
| HMSL | -13.4 | -5.8 | 1.8 | 2.1 | VEZI | -5.3 | 2.1 | 0.1 | 0.1 |
| HYMN | -20.9 | -2.7 | 0.1 | 0.1 | YYLA | -12.2 | -3.3 | 1.9 | 2.1 |
| IMLR | -11.5 | 1.6 | 2.3 | 2.6 | YZKV | -4.4 | 1.5 | 2.6 | 3.1 |
| IZMT | -5.0 | -2.1 | 0.1 | 0.1 | ZONG | -0.5 | -0.7 | 0.1 | 0.1 |


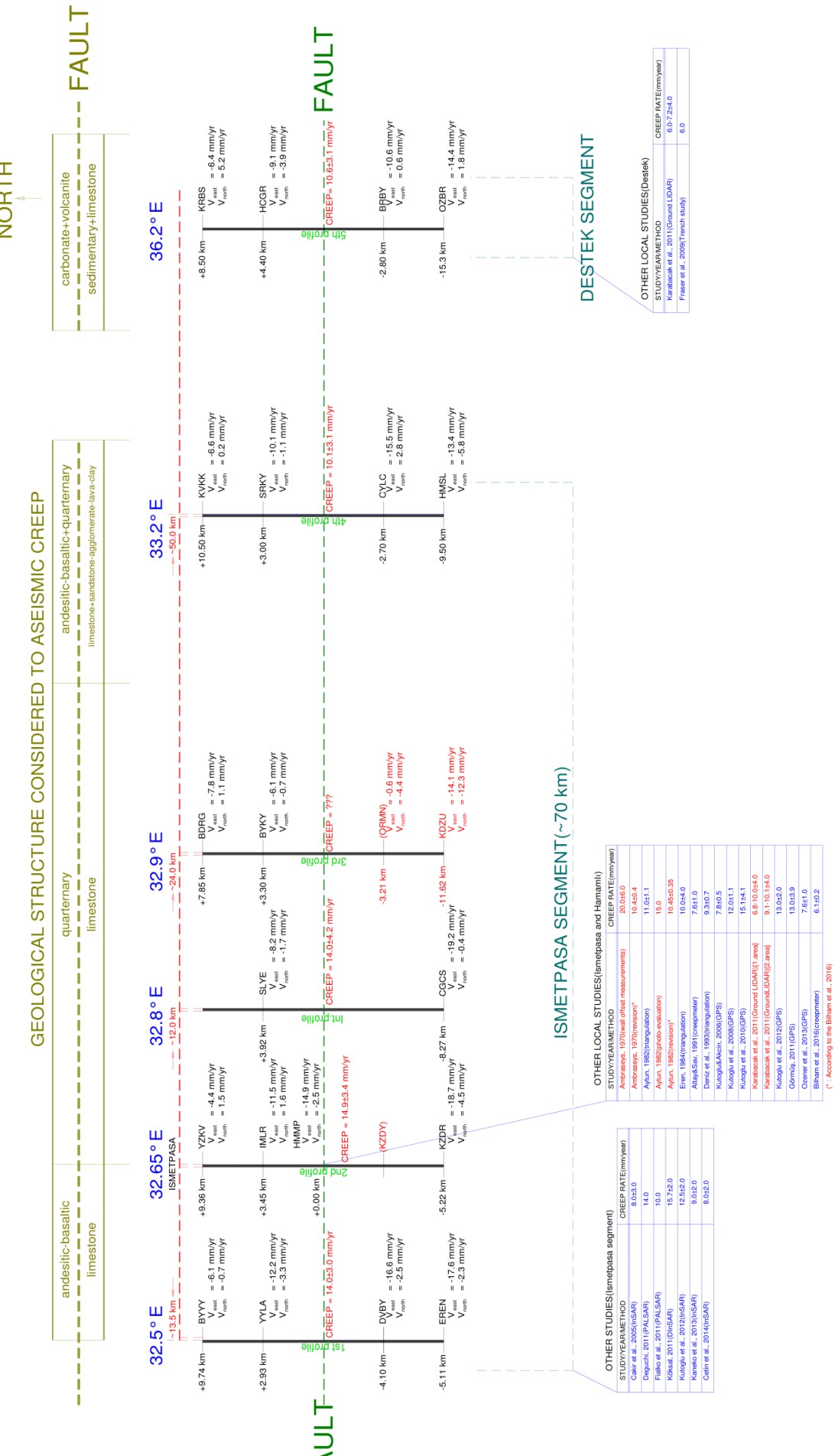

**Figure 9.** Geological structure related to the aseismic creep, station velocities, estimated creep ratio, and earlier studies around the Ismetpasa and Destek regions (Akbaş et al. 2002. , Cetin et al. 2014)

Aseismic creep ratio estimated by interpolation through the profiles using surface velocities
except the 3^rd profile at first (Table 7).
GAMIT process indicates abnormal deformation for ORMN and KDZU campaign stations, so
their data removed from the block modelling step. Additionally, the creep estimation for that profile
unfeasible. Actually, this is not a drawback for block modelling, because the remaining station
velocities are all used to model the region uneventfully.
**Table 7.** Aseismic creep rate at the Ismetpasa segment.

| Profile | Aseismic creep rate(mm/year) |
|---|---|
| 001 | 14.0±3.0 |
| 002 | 14.9±3.6 |
| 005(intermediate) | 14.0±4.0 |
| 004 | 10.1±3.0 |

With the calculated surface velocities, Destek segment also have a creep trend through the
campaign period. Estimated creep rate in this study according to GLOBK results is 10.6±3.1 mm/year
in this region, and indicates aseismic creep similar with the recent studies (Fraser et al. 2009, Karabacak
et al. 2011).
**Block Modelling**
Station velocities are suitable to predict surface and block motions around them locally. On
the other hand, observations inside the blocks provide adequate long-term block velocities and
rotations with high precision. Blocks generally demonstrate a regular movement, but their motion
differ at their boundaries from this overall velocity. They cannot move freely around the faults because
of the friction of rocks, generally infer underspeed, may down to none (Fig 10). That difference in the
velocity is called "slip deficit" and causes earthquakes after the friction threshold is surpassed (Kutoglu
and Akcin 2006, McCaffrey 2014, Yavasoglu et al. 2015).

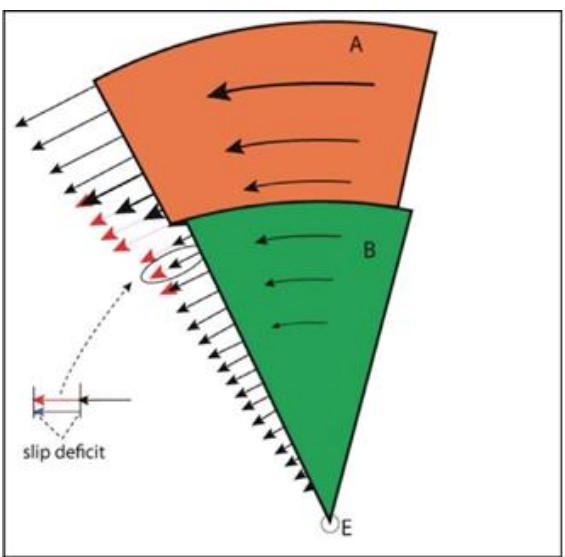


**Figure 10.** Motions of tectonic blocks around the same Euler pole and slip deficit at their boundaries. Long-term block velocities evolve at the fault zones and gap between them is responsible for strain accumulation and earthquakes (from Cakmak 2010).

Slip deficit represents that expected velocities of the blocks pass through some deformations regarding the geological structure when approaching the fault zone and frequently decreases. This is based on the geometry of the fault plane, which can only be predicted and based on the surface velocities. In that context, TDEFNODE software used in this study to predict the fault plane locking interaction regarding the depths, which calculates variations of the block motions, strain accumulation within the blocks and rotations through interseismic or coseismic period (Okada 1985, McCaffrey 2009, Yavaşoğlu 2011).

Basic input for the software includes GPS velocities, blocks with Euler poles, fault geometry and locking depth. Interacting blocks are represented as elastic blocks and assumed to have elastic deformation because of their rotation around Euler poles. All of the defined system is assumed to float inside a half-space where one of the blocks is fixed and have zero strain or movement. Fault geometry is defined by the user with nodes, and their locking ratios (phi) can be defined manually or as a function of depth (Fig. 11). Then, the software predicts the underground velocities based on the routines of *Okada* (1985) and estimates the surface velocities according the defined values. Fault geometry estimation is the key feature to minimize the difference between observed and predicted surface velocities with the help of $\chi^2$ test result, which represents the accuracy of the entire model (McCaffrey 2002, Aktuğ and Çelik 2008, Yavasoglu et al. 2011).

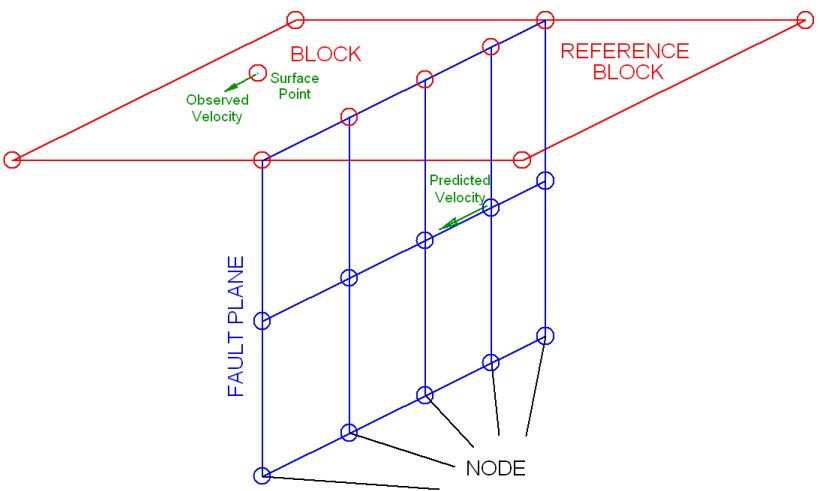

**Figure 11.** Fault plane geometry defined to the control file of TDEFNODE. Nodes divides the fault plane into sub-regions to defined depths and their locking ratio may differ from each other.

TDEFNODE is not only used for interacting blocks for interseismic strain accumulation, but also for faults which are partially or fully free slipping like aseismic creep. Software's model is suitable to define the locking ratios of all nodes independently from (0-1). (0) represents that the fault at that node is freely slipping, and (1) for a fully locked node. That allows user to define the fault plane with layers by using depth contours and to predict the fault plane if those layers are partially or fully locked (Url-2).

Aseismic creep is an earthquake-free motion along the earth surface, but in some cases it's hard to detect whether this motion is a free slipping event or an interseismic movement. Thus, the observation network around the fault plane should be planned carefully regarding the ±3-10 km station locations mentioned before (Fig 12).

During TDEFNODE process, one of the tectonic blocks should be chosen as fixed to estimate the fault parameters. Therefore, Euler pole is defined as (0, 0, 0) for the Eurasian plate and (30.7, 32.6, 1.2) for the Anatolian plate. Values represent latitude, longitude and angular velocity, respectively (McClusky et al. 2000).

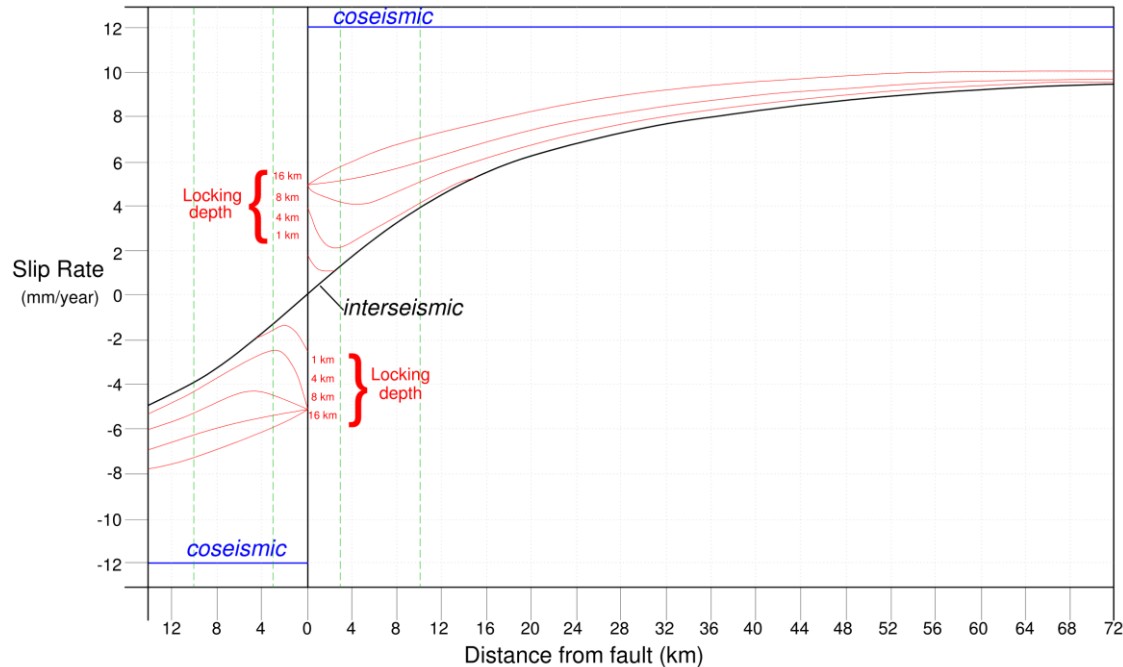

**Figure 12**. Slip rate along a fault plane during interseismic and coseismic events. Blue lines represents the coseismic, and black line represents the interseismic behaviour, where red lines demonstrates the aseismic creep ratios at two sides of the fault for different locking depths. Green lines indicates 3 and 10 km on the both side of the fault where the interseismic behavior disintegrates from aseismic creep (after Yavasoglu et al. 2015).

*Figure 12* demonstrates the suitable distances to detect aseismic creep. If an aseismic creep is suspected on a fault plane, then the optimum locations for the observation stations should be around 3 and 10 km on both sides of the fault and can be resolved from the interseismic movements. Therefore, observation stations, which are mentioned before, are established around the fault as profiles to detect this discrepancies and to detect the main locking depth of the fault and attenuation depths for the creep event. Their locations are suitable to evaluate both creeping ratios and locking depths of the faults.

**Discussion**

Station velocities all around the region indicate the relative motion of the Anatolian plate regarding the Eurasian plate. Movements ranges between 15-24 mm/year inside the southern plate where the northern motion reaches down to ~1 mm/year. That result is consistent with the previous studies (~24±2 mm/year)(McClusky et al. 2000, Reilinger et al. 2006, Yavasoglu et al. 2011). In addition, model locking depths and results are similar with a more recent study with InSAR, which indicates that the locking depth of the fault at Ismetpasa segment around 13-17 km and long-term tectonic movement is about 24-30 mm/year (Hussain et al. 2018).

Special features of the inspected segments are revealed by the network established near the
fault plane. Regarding the surface velocities of the observation points, profiles on both Ismetpasa and
Destek segments indicate movements. That ranges between 10.1-14.9 mm/year and 10.6 mm/year for
Ismetpasa and Destek segments, respectively.
Additionally, modeled fault plane evaluation for observed and calculated station movements
demonstrates similar results with the locking depths of both creeping and seismogenic layers (Fig. 13).
Station velocities on the south of the NAF are faster than the north-end as expected (Fig. 14). Regarding
the long-term geodetic block motions, modeled weighted locking ratios indicate a 13.0±3.3 mm/year
of aseismic creep all over the Ismetpasa segment. That movement does not include the whole fault
plane, thus the creeping layer seems to slip freely to 4.5 km depths from the surface and decays
between 4.5-6.75 km. The seismic data and previous studies (Cakir et al. 2005, Yavaşoğlu et al. 2011,
Hussain et al. 2019) indicate that the locking depth all over the fault as ~15 km. This result
demonstrates the fully locked portion of the fault plane is between 6.75-15 km, which supported by
the $\chi^2$ test result (1.00).


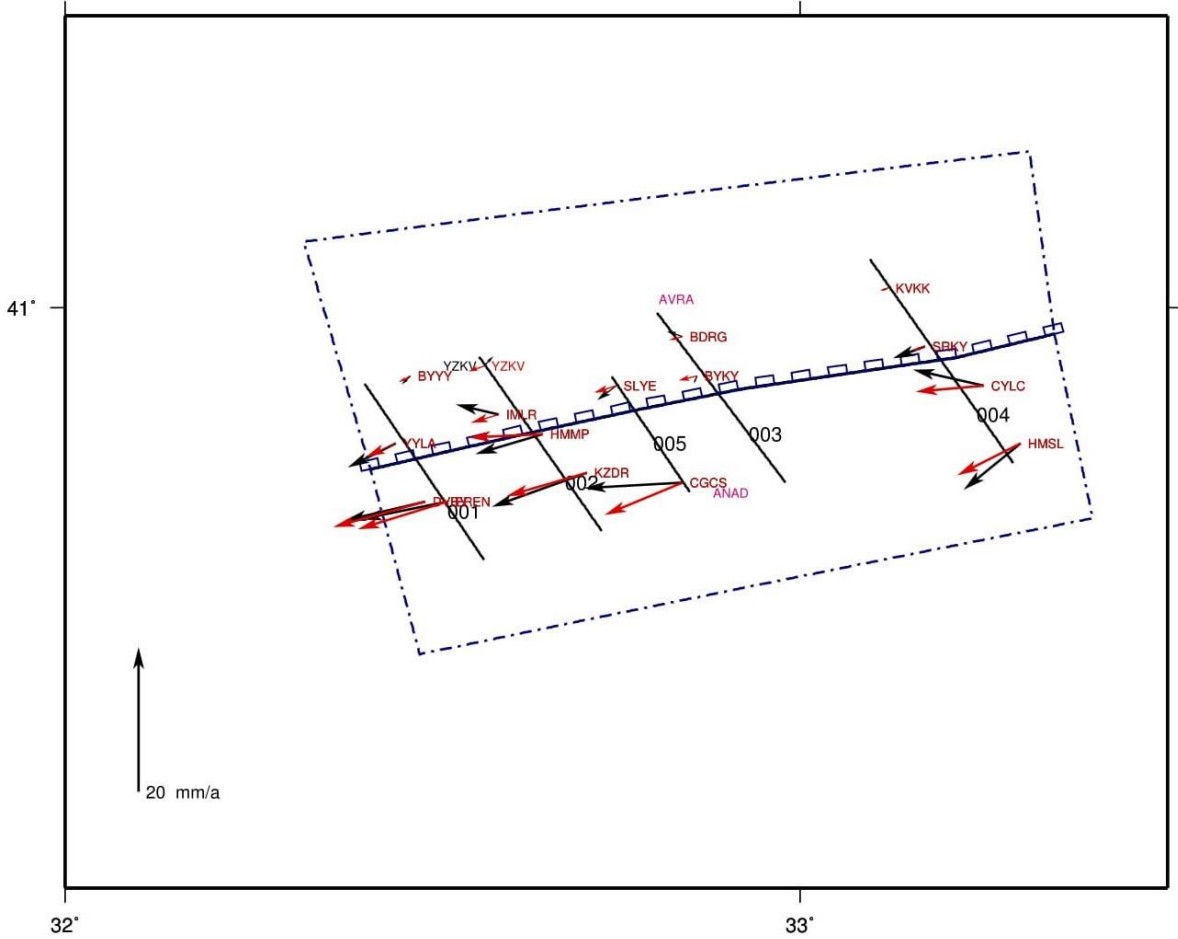


**Figure 13.** Model area for Ismetpasa segment with Eurasian plate (AVRA) on the north and Anatolian
plate (ANAD) on the south (dashed lines), divided by the creeping segment of the NAF. Black and red
arrows represent the observed and modeled velocities respectively, obtained from GAMIT/GLOBK
and TDEFNODE. Five profiles are numbered from west to east with 001 to 004, where 005 represents
the intermediate profile established during the 1st campaign. Two stations (SLYE and CGCS) on the
south-end of the profile 003 removed from the model due to unexpected velocities. Rectangles imply
the fault trace.


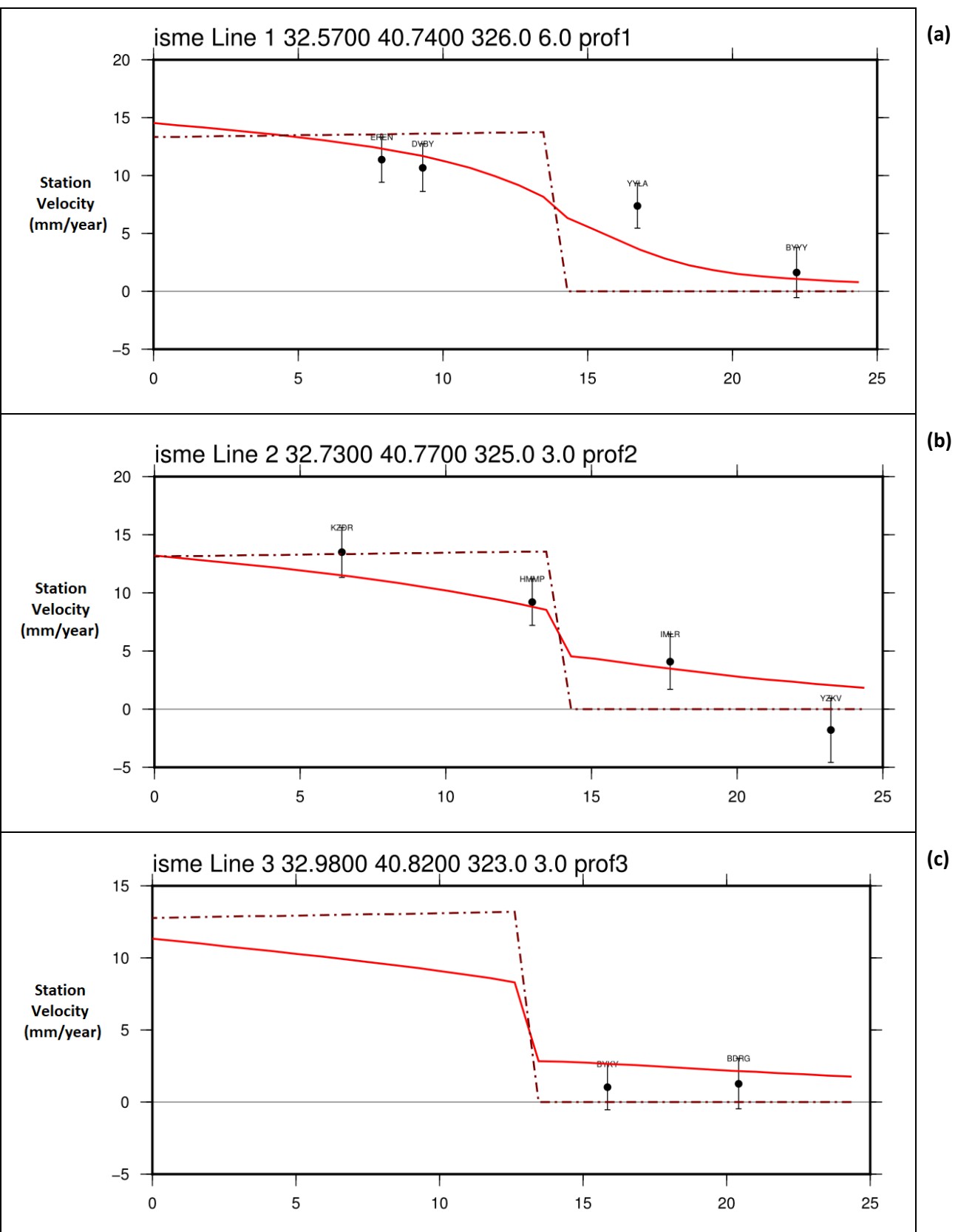

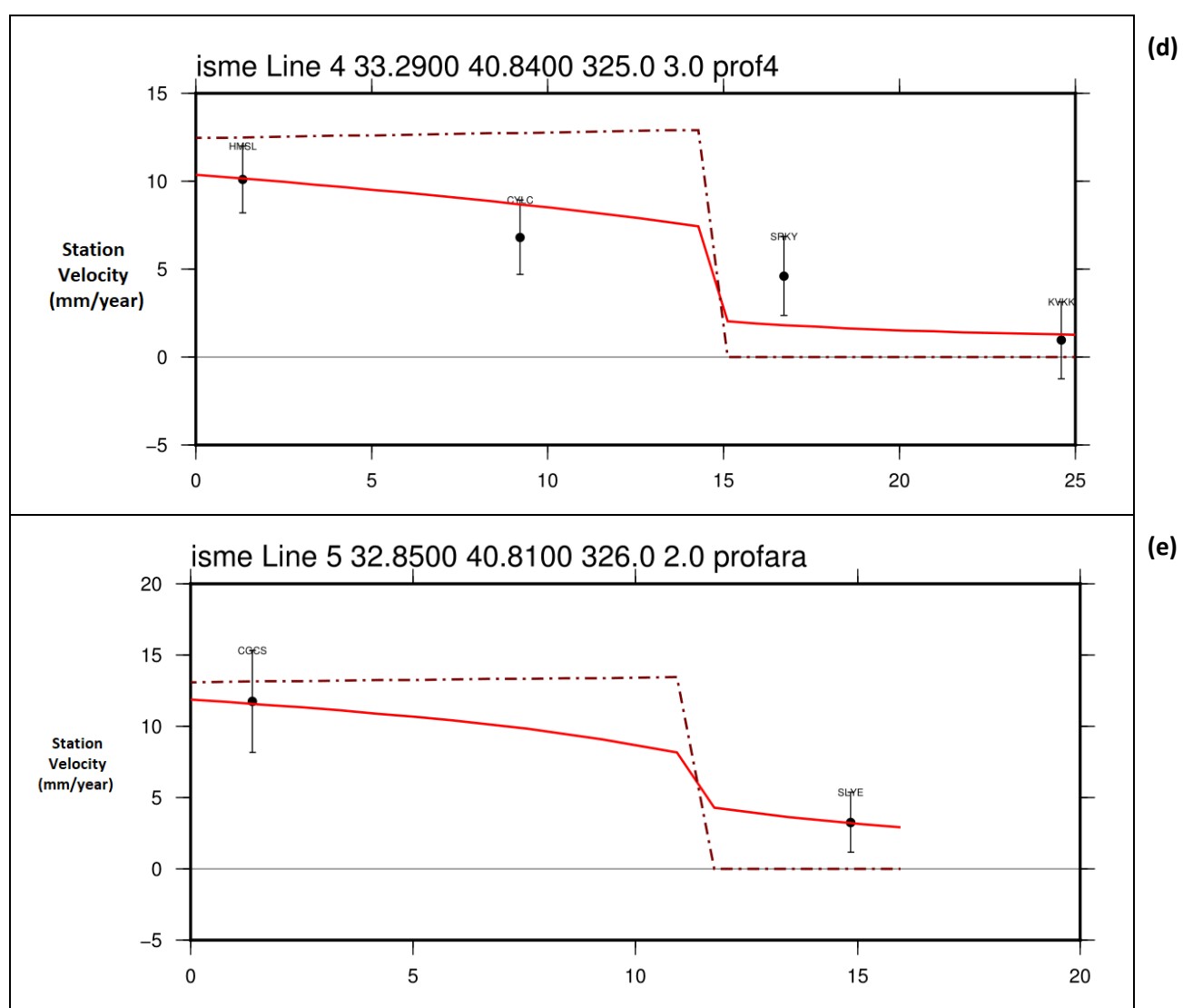

**Figure 14.** Station velocities distant 25 km for each side(east-west)through the profiles 001-005. Each station represented by a block dot, its code, and error ratio with vertical lines. Dashed lines are the block boundaries and red lines for the trend of velocity variations. Profiles 001-004 shown with a, b, c, and d, respectively. Intermediate profile(005) shown as (e). All the profiles are dispread from south to north.

Destek segment also have similar results for the observed and modeled velocities (Fig 15). The surface velocities for the profile (006) at this region indicates velocity differences (Fig 16). In addition, the modeled fault plane indicates that the creeping segment is limited to 4.3 km depth from the surface decays linearly between 4.3-6.0 km. The remaining layer of the fault seems to be fully locked down to the seismogenic layer. Free slipping portion have a 9.6 mm/year motion which is similar with the estimated surface velocities (10.6±3.1 mm/year). The $\chi^2$ test result (1.01) and the seismic data confirms the accuracy of the model.

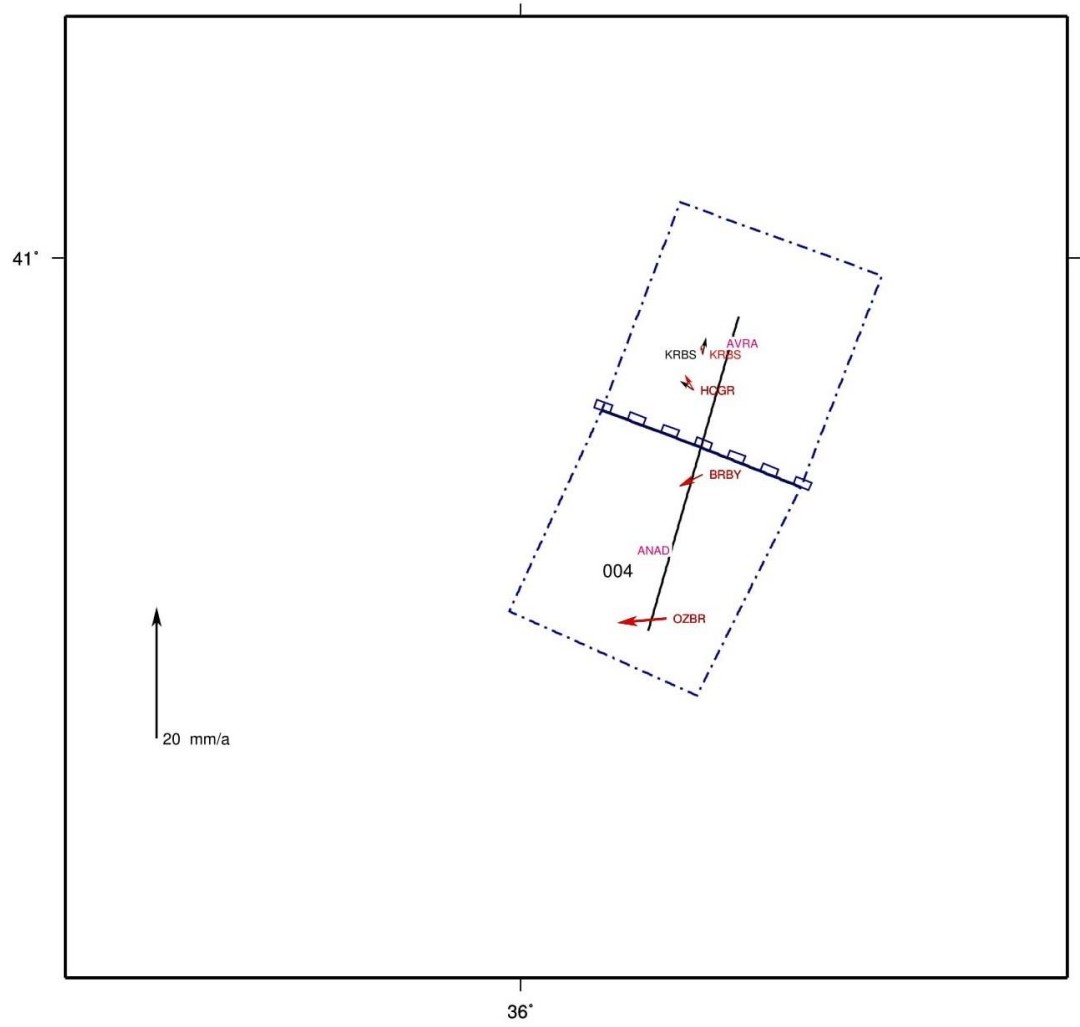


**Figure 15.** Model area for Destek segment with Eurasian plate(AVRA) on the north and Anatolian
plate(ANAD) on the south(dashed lines), divided by the creeping segment of the NAF. Black and red
arrows represent the observed and modeled velocities respectively, obtained from GAMIT/GLOBK
and TDEFNODE. 004 represents the profile in the area and rectangles imply the fault trace.

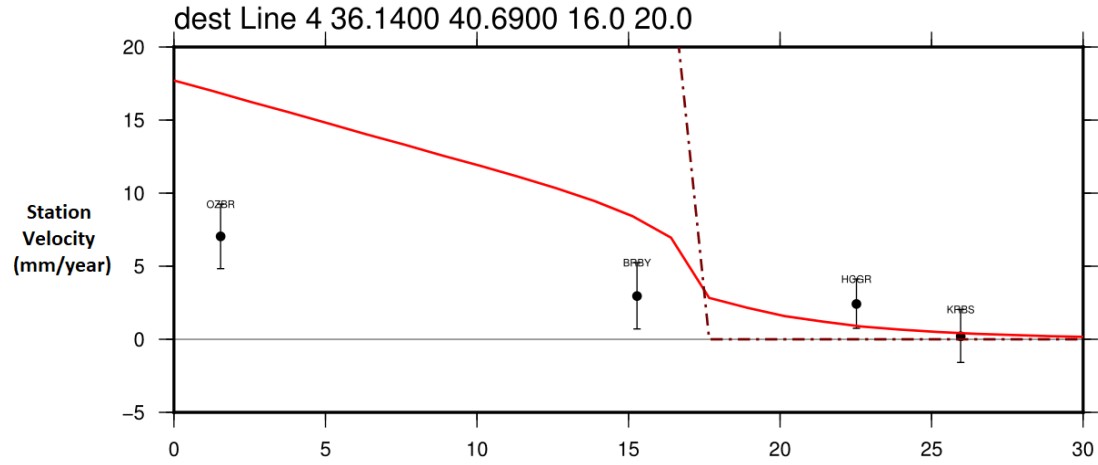


**Figure 16.** Station velocities and profile (006) for the Destek profile. Each station represented by a
block dot, its code, and error ratio with vertical lines. Dashed lines are the block boundaries, and red
lines for the trend of velocity variations. Profile dispread from south to the north.
Moreover, paleomagnetic data indicates a predominantly clockwise rotation of the blocks
bordered by the faults between Ismetpasa and Destek segments. Examining the results with this study
promotes that behavior with the GPS field of the region, especially on the Anatolian side of the NAF
(Figure 13&15) (İşseven and Tüysüz, 2006).
We find no clear evidence for attenuation at both segments. On the contrary, there is a slight
increase at Ismetpasa and almost 50% of an increase at Destek regarding the previous studies. The
frequency of this phenomenon at both segment is unclear, but results at *Hussain et al. (2018)* assists
that argument that the creep event will continue until the next large-scale earthquake.
**Conclusion**
NAF reported to have a creeping phenomena at Ismetpasa since 1970 and observed with
different techniques for a long time period with a recent discovery at Destek. All the previous studies
concentrate on whole segments or at least some regions along those segments. With this study, a GPS
network covering the whole Anatolian region along the NAF is established for the first time and results
for the velocity area used as input for block modeling. Also, the first GPS network covering Destek
segment established during this study.
Network design and location of the observation points distinguished according to the main
locking depth of the NAF and attenuation depth for the aseismic creep event. Model results show
similar outcomes for both Ismetpasa and Destek segments, where locking depth for those segments
are ~15 km, and attenuation for the creeping layer depths varies between ~4-6 km.
Through all the models, results for this study indicate that the creeping behaviour still
continues at both Ismetpasa and Destek segments, with a ratio of 13.0±3.3  mm/year and 10.6±3.1
mm/year, respectively. Block modeling and seismic data indicate that the creeping segment does not
reach to the bottom of the seismogenic layer (~15 km) and is limited to some depths, which may not
prevent a medium-large scale earthquake in the long term. In addition, we found no evidence for the
attenuation of aseismic creep. Also, the frequency of this movement at Ismetpasa is unclear and it is
not possible to predict the aseismic creep ratio precisely for long-term, but results might indicate a
small increase in the trend regarding the previous studies in the region.
Additionally, the creeping ratio seems to increase almost 50% at the Destek segment
considering the previous studies, which might indicate a relief at that segment. However, according to
the model, aseismic creep is limited to some depths (~6.0 km) and creep ratio is smaller than the long
term block movements. The increasing trend is not sufficient to release all the strain in that segment.
This might indicate strain accumulation on the both ends of the segment.

The established network by this study should be monitored periodically for the assessment of

the frequency of aseismic creep precisely, which may include possible clues for a clear fault plane
definition and earthquakes. In addition, results indicate that this creep event will be monitored to the
next earthquake, which might reveal valuable information for fault zone layout model.
**Acknowledgements**

This paper is based on the PhD thesis of Mehmet Nurullah Alkan with the title of "Kuzey

Anadolu Fayı (KAF) Bolu-Çankırı ve Amasya Bölgelerindeki Asismik Tektonik Yapının Periyodik GPS
Ölçümleri ile Belirlenmesi (Determination of Aseismic Tectonic Structure in Bolu-Çankırı and Amasya
Regions Through the North Anatolian Fault (NAF) with Periodic GPS Observations)", and supported by
the Coordinatorship of Scientific Research Projects (BAP) of Hitit University (Project No:
MYO19001.14.001), Istanbul Technical University (Project ID: 425, Code: 38146), and Afyon Kocatepe
University (Project No: 38146). We also would like to thank to all participants in this project who helped
during the field and software processes. In addition, we appreciate the great equipment and software
support of Afyon Kocatepe University and Ibrahim Tiryakioglu. The maps in this paper created by using
the GMT scripts provided by TDEFNODE software (McCaffrey 2002, URL-2, Wessel and Smith, 1995).

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
