# Peer review of "Monitoring aseismic creep trend in Ismetpasa and Destek segments throughout the NAF with a"

_Geoscientific Instrumentation, Methods and Data Systems, 2019_

## Referee Comment (RC1) · Anonymous Referee #1 · 21 Oct 2019

Yavasoglu et al. propose here a study about an interesting phenomenon that occurs along several segments of the North Anatolian Fault (NAF) : rather than being locked and generating accumulation of elastic deformation in the crust, some of them experience silent slip (creep) instead. The study focused on two sections of the NAF: one very well known, the Ismetpasa section, and a second one less known, the Destek section. A better understanding of the role of the creeping sections in the seismic cycle is definitely critical along a fault which has the potential to generate destructive earthquakes as observed in the past (for example, 1943, Mw7.2; 1944, Mw7.2; 1951, Mw6.9 in the Ismetpasa sections). It is also a good thing that Turkish scientists are working on such a topic, developing an observation network and putting effort in measuring it

often and well. These data are much needed and will be useful for the whole community working on the NAF, and beyond on all major strike-slip major faults in the world where seismic hazard is high and creep is present. Therefore they should be published eventually. Yet, I do have several important reservations about the present study.

The first one is about the choice of the measurement technics. With one point of measurement per year, the sGPS is very efficient in measuring steady processes such as interseismic deformation. Yet, previous studies evidenced unsteady behaviour of the creeping section, with burst of creep spanning several tens of days along the Ismetpasa section (Rousset et al., 2016). Because of a very low temporal resolution, survey GPS observations cannot catch that. Even if by chance the measurements are made during a transient event, longer measurements ahead would be necessary, in order to have a precise estimation of the trend before any burst, just to be able to detect it. In order to monitor transient aseismic processes, it is necessary to integrate and combine permanent continuous observations.

Second, the authors argue that the configuration of the network they built, shaping profiles with one station at 3km from the fault and one at 10km, is appropriate to monitor the creep. They are correct writing that it is "always related to the geological characteristics and fault geometry", however, I have major concerns about the ability of deciphering between models of slip with measurements so sparse and actually so far away from the fault trace that this network geometry provides.

The authors show on fig. 9 the creep rate profiles. There's a first issue, the axis is labelled "slip rate" with "mm" units. . . is it mm/yr or is it "slip" that is showed ? Let's assume it is indeed slip rate in mm/yr. The authors write "Figure 9 demonstrates the suitable distances to detect aseismic creep. If an aseismic creep suspected on a fault plane, then the optimum locations for the observation stations should be around 3 and 10 km on both sides of the fault, and can be resolved from the interseismic movements." (l.235-237) (The location of stations at 3 km and 10km on this graph could be highlighted in order to emphasize their point). More importantly, this graph does not

demonstrate the point : - The interseismic deformation is non-unique, it also depends on the locking depth and can show a strong gradient a short distance from the fault. This has to be accounted for in this graph and discussed in the text. - Creep also results in a strong gradient of surface deformation at short distance from the fault, which is showed on graph 9. At 3 km from the fault, there's a difference of 1 mm/yr between the curves of locking depth of 1 km and 4km; there's less than 2 mm/yr of differences between the 3 curves of locking depth of 8 to 16 km. It is not possible, with velocities estimated on 3 measurements over 3 years, to discriminate between these curves, which is actually implied with the "uncertainties" given in table 6, ranging between 1.8 and 2.3 mm/yr for campaign sites. It might possible to discriminate between shallow creep rate (between 0 and 5 km of locking depth) and deeper creep rate (between 5 and 15 km of locking depth).

Going further, I have concerns about the method of creep rate estimation. Maybe I did not understood it well but it is seems to me that a simple interpolation between measurements at 3 and 10 km is not good enough. Fig.6 shows the offset at the fault, which cannot reproduced by such a simple interpolation. More data at very small scale around the fault appear necessary, for example InSAR or directly surface measurements (offset sidewalk or walls as mentioned in the text l.64).

All of these lead me to seriously question the results as they are presented, for example, in figures 11 and 13. More precisely:

- Fig 11c : there are no data on the first 12 km, meaning on side of the fault according to the model, on what is based this model ?

- Fig 11d : it is in fact possible to draw a single straight line crossing all the points, same question, on what is based the model ?

- Fig 13 : same question, the model is not at all crossing the points on the south side of the fault. . .

Furthermore, the paper does need a lot of work with regards to English language usage to make it readable and understandable by the international scientific community, with recurrent grammar and conjugation mistakes (see details below).

All of these make the paper very hard to understand. Being a non-native English speaker myself, I do realize how difficult this exercise is, but it should not be the reviewer's burden and I strongly suggest that the authors have a native English speaker help with the manuscript writing before re-submitting.

Finally, most of the figures are (too) little informative or badly presented (see below more details), which seriously impact the quality of the paper. Some figures (for ex 11 and 13) are actually not commented or even called in the text, either they are useful for the study and they should be discussed or they are not and they must be removed.

For all these reasons, although I strongly encourage the authors to publish their data, I believe the present paper needs serious work, both on the methodology and on the presentation before being reconsidered for publication.

———

Additional comments:

–

* about the language:

Missing conjugated verbs for the use of past tense (not exhaustive list):

- l.93-94: "Designing a monitoring network around tectonic structures always IS related to the geological characteristics. . . "

- l.157: " . . .IGS stations WERE downloaded to cover every six month. . . "

Errors of prepositions (not exhaustive list):

- l.64: "at 1970" -> IN 1970. same l.69: "at 2003"

- l.81-83 "at Ismetpasa" / "at Destek town" -> In

Issues with the use of "THE" or not , for example (and not only) "The NAF" and not "NAF" everywhere on the paper

The formulation, used several times in the manuscript, for ex. "NAF reported to have segments which shows aseismic creep until 1970: Ismetpasa and Destek, where the second site is a more recent discovery (Ambraseys 1970, Karabacak et al. 2011)" (l.61) is incorrect, it is not the NAF which reports, it is Ambraseys 1970 and Karabacak 2011, eg "previous studies reported segments . . .".

——

* About the figures:

- Figure 1: it is useful to have a first context figure but it does not seem useful to show it at such a large scale, it could be centered on the NAF between 23 and 40°E, 35 and 42°N. I guess everything that is not mentioned in the text, therefore that does not have an influence on the creeping segments, should not need to be on the figure. On the contrary, it misses quite a lot of important information for the understandings of the paper: the very first one being where are the locations of Ismetpasa and Destek ? Please, more generally, show on the map ALL the location of cities mentioned in the text (Baymoren & Gerede for ex.)? The authors also mentioned the historical seismicity along the 2 segments (l.61-68), where did these earthquakes occur exactly respective to the 2 creeping segments? The GPS network at this scale would also be interesting to actually have a sense of its footprint.

- Figure 2: the label of seismogenic zone is missing.

- Figure 3: I don't really see the point of this figure, the scale is too small to be able to locate the region on the NAF, and it is too large to see any hints of aseismic creep ? Is there any pictures showing the creep ? If so, they could be added as a composite figure showing this pictures and their location ? As it is, this figure is useless.

[Figure]

- Figure 4: On sub-figure (b), there are 3 fault trace, the GPS profile only encompasses 2 of them... where is supposed to occur the creep ? Why ignoring the 3rd fault? Discuss that.

- Figure 5: the dataset is quite dense which make this figure difficult to read. Typically, it is hardly possible to read the station codes - which are in fact not needed. There again, rescale the map : there are no data from 26 to 28°E and from 38.1 to 40°E. The uncertainty is missing from the arrow legend. The fault trace, even simplified, should appear. Caption: "relative to fixed Eurasia" instead of "when Eurasian plate selected as fixed". Later on: "the westward motion of the Anatolian plate" instead of "the Anatolian plate's motion to the west"

- Figure 6: this figure is very complicated and I am not sure it is really useful. The geological structure is hardly mentioned in the text, and the creep values estimated in previous studies are already recapitulated in table 1. Table 7 and table 1 could be gathered, ordering table 1 as function of profiles and then adding the creep values from this study to compare them ?

- Figure 7: this is one is directly taken from a PhD unmodified, maybe it can go in supplement ?

- Figure 10 - 12: why all the white on these 2 figures instead of zooming in on the data ? Add red arrow and the legend "model" / "observed" along with the scale. What are all the squares lying on the fault ?

- Figure 11 - 13 : there's obviously no data further than 25 km from the fault, re-scale the profiles. Same remark for the y-axis, the smallest velocity is -2 or -3 mm/yr, re-scale the velocity axis. What are the dashed red lines ? What is the "transverse"?

———

*About the GPS data:

Missing info:

- coordinates of the Euler pole estimated to rotate the velocities in fixed Eurasia.

- coordinates of all sites (Table 6). In which frame are given the velocities ? ITRF08 or fixed Eurasia ? Indicate it but velocities both in ITRF08 and fixed Eurasia should be given.

- could be useful to have a table showing the date of sGPS measurements:

_______________________________________________________-

| STAT | 235 236 237 238 241 | ......

| BYYY | X X - -

.......

It will allow to better evaluate the double difference process (double difference process implies to have simultaneous measurements, in order to estimate baselines.)

Table 5 could be gathered with table 4 with a symbol with stations used for stabilitation.

l.159-160: "GPS data for cGPS and IGS stations downloaded to cover every six month between August 2009-2016 to increase the stabilisation at the GLOBK step." I don't understand what means "to cover every 6 months" ? The stabilization is important over the campaign dates, then if the stabilisation stations are IGS stations, their positions and velocities are very well known in the ITRF08 . . . Another robust stabilization approach, maybe more efficient than processing data over a longer period than the campaign, is to combine IGS h-files at the dates of campaign in the GLOBK process (to download either from SOPAC or MIT – code sh_get_hfiles in gg).

l.165-167: "Results show that the velocity of the stations inside the Anatolian plate are gathering up to 15- 20 mm/year (Fig 5), which is similar with the previous studies (McClusky et al. 2000, Reilinger et al. 2006, Yavaşoglu et al. 2011)." In what frame ? ITRF08 or fixed Eurasia ? What does mean Ấninside the Anatolian plateÂż ? Located on the Anatolian plate ? "ranging from 15 to 20 mm/yr" instead of "gathering up to. . . "

Tables 6 : The uncertainties given in table 6 (of less than 0.1mm/yr in some cases) are totally unrealistic, they must be formal errors from the globk process, in which case it is necessary to precise at how many sigmas. Going further I think the authors are mixing "errors", "uncertainties" and "repeatabilities" (l.179 : "repeatability of the ORMN and KDZU stations indicate abnormal deformation"). They are different things, please clarify what is used, and indicate all the necessary information.

I have a printed annotated manuscript with additional comments that I can provide if requested by the authors.

---

## Referee Comment (RC2) · Anonymous Referee #2 · 5 Nov 2019

REVIEW GI-2019-24 Monitoring aseismic creep trend in Ismetpasa and Destek segments throughout the NAF with a Ââǎlarge scale GPS network Summary In this study, the authors established GNSS network Ismetpasa and Destek segments on the NAF (North Anatolian Fault). GNSS measurements were used to understand the tectonic mechanism of these segments and three campaign data were collected in 2014, 2015 and 2016. GNSS data has been evaluated by GAMIT / GLOBK software. Tectonic modeling has been carried out after GNSS evaluation. For this, they used TDEFNODE. As a result of modeling using geodetic data, it was tried to reveal the tectonic mechanism of these segments. They calculated surface velocities aseismic creep still continues to some rates, ∼13 mm/year at Ismetpasa and ∼10 mm/year

at Destek. Also, they obtained between ∼4-6 km for the creep depth layers. The manuscript is very interesting. Introduction of field and problem as well used method are well written. This manuscript will make a significant contribution to the study of earth sciences. Therefore, I think that this publication is suitable for the concept of this journal and should be printed after minor revisions are made. Some of my comments and suggested corrections are given in detail below; • The abstract can be extended with the results of block geometry. • There are many fault names mentioned in the paper. The fault name should be provided in Figure 4. • The previous results of the studies conducted to determine creep rate in the Ãřsmetpaşa segment between 1970 and 2016 can be given as figure. • Some of content is repeated, For example Page 7 last paragraphÂǎ (line 143-147) same as 'GPS Data Evaluation' section line 158-160. Suggest revising or deleting. • There are Turkish sentences or word in Figure 7 and in the manuscript. These sentences should be deleted in the paper. • The citation publications and references should be checked, eg; Poyraz vd. 2011 instead of Poyraz et al. 2011. • Some figures are not enough resolution. So, these figures should be rearranged. For example, Figure 5,6 and 11. • The parameter values used in block modeling (such as locking depth, Euler poles) should be explained in a few sentences in the paper. • The chi-square value can be given in text (between lines 254-263 in the page 16). It will be more meaningful.

Please also note the supplement to this comment:
https://www.geosci-instrum-method-data-syst-discuss.net/gi-2019-24/gi-2019-24-RC2-supplement.pdf

---

## Author Comment (AC1) · 4 Dec 2019

**RESPONSE FOR ANONYMOUS REFEREE #1:**

**COMMENT #1 (PAGE 2):**

"...Because of a very low temporal resolution, survey GPS observations cannot catch that. Even if by chance the measurements are made during a transient event, longer measurements ahead would be necessary, in order to have a precise estimation of the trend before any burst, just to be able to detect it. In order to monitor transient aseismic processes, it is necessary to integrate and combine permanent continuous observations..."

**Response:**

Tectonic movements, like aseismic creep, can be monitored even using long-term campaign observations. Slip deficit is the key factor to determine if creep exists or not. In that case, it's not an essential issue to establish permanent stations. Results would lead us for this kind of permanent continuous observations if necessary. Also, earlier studies which uses space geodesy didn't require or mention permanent GPS stations for this phenomena, and final outcomes of these studies given at Table 1&2 and Figure 6.

**COMMENT #2 (PAGE 2):**

"...They are correct writing that it is "always related to the geological characteristics and fault geometry", however, I have major concerns about the ability of deciphering between models of slip with measurements so sparse and actually so far away from the fault trace that this network geometry provides."

**Response:**

Figure 9 includes *Yavasoglu et al. 2015* graphics that shows the optimum perpendicular distances from a creeping fault, 3 and 10 km on both sides. Project mainly maintained on this basis to configure profiles and yearly observations. We try to understand block movements around the region, and results of the TDEFNODE modeling indicates that the distribution of the stations were sufficient to represent blocks along the creeping parts of the NAF.

**COMMENT #3 (PAGE 2):**

"The authors show on fig. 9 the creep rate profiles. There's a first issue, the axis is labelled "slip rate" with "mm" units: : : is it mm/yr or is it "slip" that is showed?..."

**Response:**

Figure fixed as "mm/year" for the axis.

**COMMENT #4 (PAGE 2):**

"...(The location of stations at 3 km and 10km on this graph could be highlighted in order to emphasize their point)..."

**Response:**

Figure 9 revised as follows:

**Figure 9**. Slip rate along a fault plane during interseismic and coseismic events. Blue lines represents the coseismic, and black line represents the interseismic behaviour, where red lines demonstrates the aseismic creep ratios at two sides of the fault for different locking depths. Vertical green lines indicates 3 and 10 km on the both sides of the fault where the interseismic behaviour disintegrates from aseismic creep (after Yavasoglu et al. 2015).

**COMMENT #5 (PAGE 3):**

"...- The interseismic deformation is non-unique, it also depends on the locking depth and can show a strong gradient a short distance from the fault. This has to be accounted for in this graph and discussed in the text..."

**Response:**

By this project, we established GPS networks around the both regions, Ismetpasa and Destek. This gives us the opportunity to monitor a large area. For this reason, several campaign stations established around the NAF to represent the block movements, which based on the theoretical studies.

**COMMENT #6 (PAGE 3):**

"...Fig.6 shows the offset at the fault, which cannot reproduced by such a simple interpolation. More data at very small scale around the fault appear necessary, for example InSAR or directly surface measurements (offset sidewalk or walls as mentioned in the text I.64)..."

**Response:**

This project based on GPS observations. InSAR or direct measurements on the field and involving these data with our results is another research issue for the future. Interpolation along the profiles from the GAMIT/GLOBK results gives us a quick overview for the creep behavior, they are not used for final outcomes.

Also, we didn't get any result about the creep rate at the 3rd profile, because it was impossible to estimate the movement using interpolation due to local deformation at the south of the profile, and station velocities removed from the input data used to model the fault and blocks with TDEFNODE. This procedure explained in the text.

**COMMENT #7 (PAGE 3):**

"...- Fig 11c : there are no data on the first 12 km, meaning on side of the fault according to the model, on what is based this model ?

- Fig 11d : it is in fact possible to draw a single straight line crossing all the points, same question, on what is based the model ?

- Fig 13 : same question, the model is not at all crossing the points on the south side of the fault..."

**Response:**

Our model based on Figure 9 elementarily but there are some limitations when applied on the field. Also, those fault perpendicular distances are not the exact locations to seize creep; they should be around those locations.

Another issue is that the site selection is heavily relevant with the ground truth. It was not always possible to find out a suitable location for campaign points at the given distances/locations, or they cannot maintain a straight profile on practical applications (inconvenient soil structure, impractical locations for GPS observations due to surrounding obstacles, etc.). For these reasons, we select the closest locations for the stations based on our model.

**COMMENT #8 (PAGE 4):**

"...Furthermore, the paper does need a lot of work with regards to English language usage to make it readable and understandable by the international scientific community, with recurrent grammar and conjugation mistakes (see details below)."

All of these make the paper very hard to understand. Being a non-native English speaker myself, I do realize how difficult this exercise is, but it should not be the reviewer's burden and I strongly suggest that the authors have a native English speaker help with the manuscript writing before re-submitting..."

**Response:**

Based on this comment, a total check including proofreading has done.

**COMMENT #9 (PAGE 5):**

"...Figure 1: it is useful to have a first context figure but it does not seem useful to show it at such a large scale, it could be centered on the NAF between 23 and 40°E, 35 and 42°N. I guess everything that is not mentioned in the text, therefore that does not have an influence on the creeping segments, should not need to be on the figure. On the contrary, it misses quite a lot of important information for the understandings of the paper: the very first one being where are the locations of Ismetpasa and Destek ?

*Please, more generally, show on the map ALL the location of cities mentioned in the text (Baymoren & Gerede for ex.)?..."*

**Response:**

Following figure prepared for the manuscript. Both segments have their labels according to the nearest villages, thus İsmetpaşa and Destek settlements shown on the figure.

**Figure 8.** Active fault segments on the North Anatolian Fault (NAF). Blue rectangles defines İsmetpaşa and Destek segments from west to east, respectively (after Bohnhoff et al. 2016).

**COMMENT #10 (PAGE 5):**

"...The authors also mentioned the historical seismicity along the 2 segments (l.61-68), where did these earthquakes occur exactly respective to the 2 creeping segments? The GPS network at this scale would also be interesting to actually have a sense of its footprint..."

**Response:**

Following figure added in the manuscript.

---

## Author Comment (AC2) · 4 Dec 2019

**RESPONSE FOR ANONYMOUS REFEREE #2:**

*COMMENT #1 (PAGE 2):*

*"…The abstract can be extended with the results of block geometry…"*

**Response:**

According to this comment, following statement added to the abstract:

*"Also, aseismic creep behavior is limited to some depths and decays linearly to the bottom of seismogenic layer at both segments."*

*COMMENT #2 (PAGE 2):*

*"…There are many fault names mentioned in the paper. The fault name should be provided in Figure 4…"*

**Response:**

Profile at Destek segment established on the creeping fault trace. Faults on the south are secondary faults and no aseismic creep reported at those locations. To clarify the situation, following statement added to the explanation of Figure 4:

*"Fault traces on the south of profile 006 are secondary faults."*

*COMMENT #3 (PAGE 2):*

*"…The previous results of the studies conducted to determine creep rate in the Ismetpasa segment between1970 and 2016 can be given as figure…"*

**Response:**

Table 1 and Figure 6 includes those information. A figure would be more complex to demonstrate all the studies because they would interlace each other.

***COMMENT #4 (PAGE 2):***

*"…Some of content is repeated. For example ´Page 7 last paragraph (line 143-147) same as 'GPS Data Evaluation' section line 158-160. Suggest revising or deleting…"*

**Response:**

That repeated text deleted.

***COMMENT #5 (PAGE 2):***

*"…There are Turkish sentences or word in ´Figure 7 and in the manuscript. These sentences should be deleted in the paper…"*

**Response:**

Figure 7 and Turkish sentences fixed according to the comment.

***COMMENT #6 (PAGE 2):***

*"…The citation publications and references should be checked, eg; Poyraz vd. 2011 ´ instead of Poyraz et al. 2011…"*

**Response:**

Citations fixed in the text.

***COMMENT #7 (PAGE 2):***

*"…Some figures are not enough resolution. So, these figures should be rearranged. For example, Figure 5,6 and 11…"*

**Response:**

Figure 5 replaced with the high resolution copy.

Figure 6 has the highest resolution and prepared with another software. It is the best output of the that.

Figure 11 includes 5 different profiles. They are rearranged according to the comment.

*COMMENT #8 (PAGE 2):*

*"…The parameter values used in block modeling (such as locking depth, Euler poles) should be explained in a few sentences in the paper…"*

**Response:**

That information added in the text as follows:

*"During TDEFNODE process, one of the tectonic blocks should be chosen as fixed to estimate the fault parameters. Therefore, Euler pole defined as (0, 0, 0) for Eurasian plate and (30.7, 32.6, 1.2) for Anatolian plate. Values represent latitude, longitude and angular velocity, respectively (McClusky et al. 2000)."*

*COMMENT #9 (PAGE 2):*

*"…The chi-square value can be given in ´text (between lines 254-263 in the page 16)…"*

**Response:**

Chi-square results are (1.00) and (1.01) for Ismetpasa and Destek segments, respectively. These results added in the text before the figures of modeled area.